# Comparison of Accuracy in the Evaluation of Nutritional Labels on Commercial Ready-to-Eat Meal Boxes Between Professional Nutritionists and Chatbots

**DOI:** 10.3390/nu17193044

**Published:** 2025-09-24

**Authors:** Chin-Feng Hsuan, Yau-Jiunn Lee, Hui-Chun Hsu, Chung-Mei Ouyang, Wen-Chin Yeh, Wei-Hua Tang

**Affiliations:** 1Division of Cardiology, Department of Internal Medicine, E-Da Hospital, I-Shou University, Kaohsiung 82445, Taiwan; calvin.hsuan@msa.hinet.net; 2School of Medicine, College of Medicine, I-Shou University, Kaohsiung 82445, Taiwan; 3Division of Cardiology, Department of Internal Medicine, E-Da Dachang Hospital, I-Shou University, Kaohsiung 807066, Taiwan; 4Department of Internal Medicine, Lee’s Clinic, 130 Min-Zu Rd, Pingtung 90000, Taiwan; lee@leesclinic.org; 5Department of Nursing, Lee’s Clinic, 130 Min-Zu Rd, Pingtung 90000, Taiwan; huichun.hsu@leesclinic.org; 6Department of Clinical Nutrition, Lee’s Clinic, 130 Min-Zu Rd, Pingtung 90000, Taiwan; maggieoy@gmail.com; 7Division of Cardiology, Department of Internal Medicine, Ministry of Health and Welfare Yuli Hospital, Hualien 98142, Taiwan; 8Faculty of Medicine, School of Medicine, National Yang Ming Chiao Tung University, Taipei 112304, Taiwan

**Keywords:** artificial intelligence, nutrition estimation, large language models, ChatGPT, Claude, Grok, Copilot, Gemini, food label

## Abstract

**Background/Objectives**: As convenience store meals become a major dietary source for modern society, the reliability of their nutrition labels is increasingly scrutinized. With advances in artificial intelligence (AI), large language models (LLMs) have been explored for automated nutrition estimation. **Aim**: To evaluate the accuracy and clinical applicability of AI-assessed nutrition data by comparing outputs from five AI models with professional dietitian estimations and labeled nutrition facts. **Methods**: Eight ready-to-eat convenience store meals were analyzed. Four experienced dietitians independently estimated the meals’ calories, macronutrients, and sodium content based on measured food weights. Five AI chatbots were queried multiple times with identical input prompts to assess intra- and inter-assay variability. All results were compared to the official nutrition labels to quantify discrepancies and cross-model consistency. **Results**: Dietitian estimations showed strong internal consistency (CV < 15%), except for fat, saturated fat and sodium (CVs up to 33.3 ± 37.6%, 24.5 ± 11.7%, and 40.2 ± 30.3%, respectively). Among AI models, ChatGPT4.o showed relatively consistent calory, protein, fat, saturated fat and carbohydrate estimates (CV < 15%), and Claude3.7, Grok3, Gemini, and Copilot showed caloric and protein content as consistent (CV < 15%). Sodium values were consistently underestimated across all AI models, with CVs ranging from 20% to 70%. The accuracy of nutritional fact estimation over the five AI models for calories, protein, fat, saturated fat and carbohydrates was between 70 and 90%; when compared to the nutritional labels of RTE, the sodium content and saturated fat estimated were severely underestimated. **Conclusions**: Current AI chat models provide rapid estimates for basic nutrients and can aid public education or preliminary assessment; GPT-4 outperforms peers in calorie and potassium-related estimations but remains suboptimal in micronutrient prediction. Professional dietitian oversight remains essential for safe and personalized dietary planning.

## 1. Introduction

In recent years, the application of artificial intelligence (AI) chatbots in the field of medical nutrition has attracted significant attention. AI models leverage natural language processing to provide interactive, real-time responses, demonstrating revolutionary potential in electronic health consultations [1]. The existing literature has begun exploring the reliability of these AI systems in delivering nutritional information. For example, a study by Haman et al. showed that ChatGPT demonstrated a high level of accuracy in providing food nutrition data, with approximately 97% of its caloric estimates falling within ±40% of USDA reference values. It also showed good reproducibility, with relatively low coefficients of variation in nutrient estimates [2]. Furthermore, when tasked with generating a one-day meal plan under caloric constraints, the caloric content of meals generated by ChatGPT typically stayed within ±30% of the target values. Overall, these results suggest that ChatGPT has the potential to deliver fairly accurate and consistent nutritional information [2].

Similarly, Hoang et al. compared nutrient estimates provided by AI and dietitians for 222 food items. The results revealed no significant differences in estimates of calories, carbohydrates, and fat, though ChatGPT significantly overestimated protein content. In that study, ChatGPT estimated calorie content within ±10% of actual values for approximately 35–48% of the foods evaluated [1]. Notably, the updated ChatGPT-4 model outperformed ChatGPT-3.5, producing values closer to reality, although both versions tended to overestimate protein [2]. These findings preliminarily affirm the feasibility of using AI for dietary nutrition assessment, while also highlighting the need for further examination of biases, particularly for specific nutrients like protein.

More importantly, some AI models showed limited reproducibility and consistency in nutrition data. The same study noted that across different days, nutrition values provided by some AI systems fluctuated significantly, deviating from dietitians’ calculations by as much as 45% in caloric estimates [3]. This variability poses a major challenge for precision-required contexts such as diabetes meal planning. Therefore, further evaluation of the accuracy and stability of AI models in dietary nutrition analysis is warranted, particularly in comparison to assessments by professional dietitians and actual product labeling. Dietitians remain the professional benchmark for dietary assessment, but their estimates may vary due to hidden ingredients, preparation methods, and portion-size interpretation [4]. Thus, comparing AI systems directly with dietitians provides an essential test of real-world applicability.

In modern society, due to fast-paced lifestyles, many people have irregular eating habits. Convenience and accessibility often lead to diets dominated by high-fat, high-sugar, and high-salt foods, increasing risks of obesity, hypertension, and cardiovascular diseases, posing a significant public health challenge [5]. Hence, understanding and selecting healthy foods and maintaining balanced nutrient intake have become increasingly important. However, identifying the nutritional composition of food is a complex and time-consuming task for most individuals, often requiring ingredient knowledge, food scales, and extensive manual calculations. Simplifying this process through technologies that can predict nutrition from food images offers a practical and user-friendly solution, especially for individuals aiming to improve dietary habits. Emerging machine learning techniques, including AI, may enable automatic, accurate, and robust prediction of food properties, facilitating rapid food analysis and personalized dietary recording [6]. Ready-to-eat (RTE) convenience foods represent a particularly relevant test case, as they are widely consumed but highly variable in composition. Prior studies have shown discrepancies between labeled and actual nutrient values, particularly for calories, fat, and sodium [6,7,8]. This raises clinical concerns, as relying solely on label information may mislead patients requiring precise nutrient control, such as those with diabetes or hypertension.

These meals often feature labeled nutritional content including calories, carbohydrates, protein, and fat. However, current studies indicate that commercial labeling may not be entirely accurate. Some meals differ significantly from their labeled values [7]. For example, one analysis of U.S. restaurant meals found that actual calorie content exceeded the labeled values by an average of 18%, while frozen meals exceeded by 8% [7]. Another study found that approximately 19% of restaurant meals had at least 100 more calories than claimed [8]. This discrepancy is partly due to regulatory allowances: the U.S. FDA permits up to a 20% margin of error in calorie labeling. Similar challenges have been noted in Taiwan and other regions, underscoring the importance of validating both AI- and dietitian-derived estimates against label references [9].

Therefore, relying solely on label information to evaluate nutrition, especially for diabetic meals, may be risky and necessitate more rigorous validation. Overall, the current literature indicates that while AI shows promise for nutrition evaluation, its performance compared to professional dietitians still requires further validation. Additionally, the reliability of commercial meal labeling remains a clinical concern.

In this study, we conduct the first evaluation comparing five types of conversational AI chatbots and assess their interpretation of labeled nutrition data from pre-packaged commercial meals. We also compare these assessments with the nutrient predictions made by professional dietitians using ingredient-based quantification models. The study integrates three components—product labeling, AI chatbot predictions, and dietitian assessments—for a comprehensive comparative analysis, aiming to better understand their differences, highlight the strengths and limitations of each approach, and evaluate the feasibility of AI in nutrition assessment.

## 2. Materials and Methods

The primary aim of this study was to evaluate the precision and reproducibility of AI-assisted nutrient estimation under the assumption that labeled values were correct, rather than to validate absolute accuracy against laboratory measurements. Specifically, to assess how accurately AI can estimate key nutritional values (calories, macronutrients, sodium, fiber, etc.) from images and compare AI-generated assessments with independent evaluations conducted by registered dietitians. A comparative cross-sectional study using a dataset of commercial RTE meal photos collected from convenience store chains. This study compares nutritional composition assessments of commercial standard meals across different AI Chatbot evaluations and benchmarks against assessments made by four professional clinical dietitians.

### 2.1. Meal Sample Selection

Representative RTE meals were selected from the market, specifically commercial boxed meals from 7-Eleven. The sampled meals, made in Taiwan and obtained from 7-Eleven outlets, are classified as ultra-processed convenience foods under the NOVA system, meaning industrially formulated, RTE products [10]. The samples cover a range of meal types to reflect common dietary patterns among the general population. To control for variability, eight boxed meals were chosen: soy sauce fried noodles, high-protein muscle meal, grilled chicken &veggie meal, honey-glazed chicken leg with pork patty, thick-cut pork chop rice, turkey meal box, crispy soybean crumb baked fish and rose soy sauce chicken.

### 2.2. AI Chatbot Nutritional Facts Assessment

Five freely accessible, regularly updated AI chatbots were employed to estimate the nutrient content of the same meals: OpenAI’s ChatGPT (GPT-4o model), Google Gemini, xAI’s Grok 3, Microsoft Copilot (Edge Stable version 134.0.3124.51), and Anthropic’s Claude 3.7. All analyses were conducted between January and April 2025. High-resolution images of the RTE meals were captured and input into the AI models. To minimize portion-size underestimation, mixed foods were separated, allowing clearer recognition of components and more accurate nutrient estimation. The AI was asked (by prompt) to generate the following nutritional facts, including weight, calories (kcal), carbohydrates, protein, total fat, saturated fat, dietary fiber, and sodium contents. These AIs were also asked for their relevance in nutritional counseling and meal planning capabilities by generating the nutritional effects and possible effects on glucose and lipid levels. To standardize the analysis, a unified prompt was designed for each meal, which instructed the AI to act as a catering dietitian and analyze the meal (photo-based input) using the Taiwan Food Composition Database and nutrition labeling regulations [11]. The AI was also asked to present a table of food amounts and nutrient values and indicate possible allergens and precautions as in a previous study [1]. Each AI was prompted three times per meal to evaluate consistency. Any significantly different responses were recorded. Nutrient estimations from all AIs for each meal were compiled. Research shows that input language (Chinese vs. English) has minimal impact on ChatGPT’s results [1], so both Traditional Chinese and English were used in this study.

### 2.3. Professional Dietitian Assessment

Four registered dietitians independently estimated nutrient content following a standardized workflow: (i) deconstruction of the meals into components (starches, meats, vegetables, sauces), (ii) weighing each component, (iii) assigning food codes from the Taiwan Food Nutrition Database and the Taiwan’s Ministry of Health and Welfare Food Substitution Table [11], (iv) converting to gram equivalents and summing nutrients via spreadsheets, and (v) outlier cross-check. No proprietary or automated nutrition software was used—only official databases and exchange lists supported by MOHW. Dietitians were blinded to label values and followed predefined rules for sauces/oils (e.g., default oil uptake for stir-fry; sodium allocation for marinades).

### 2.4. Data Organization

This study compares the discrepancies between commercial label values, dietitian estimates, and AI-generated estimates. For each meal, the nutrient values estimated by AI models and dietitians were compared against the labeled values. Percentage deviation was calculated as (estimated value − labeled value) ÷ labeled value × 100, where positive values indicate overestimation and negative values indicate underestimation relative to the label. To evaluate variability among dietitians, coefficients of variation (CV%) were calculated across their independent estimates for each nutrient. Differences between AI- and dietitian-derived assessments were further summarized as mean deviations and standard deviations across all meals. In addition to percentage deviations from labeled values, inter-dietitian variability was quantified using the coefficient of variation (CV%). For each nutrient (energy, protein, fat, saturated fat, carbohydrate, sodium), the CV% was calculated across the four independent dietitian estimates for all eight RTE meals. The mean CV% values presented in Table 1 therefore summarize the degree of variation among dietitians’ assessments, providing a benchmark for comparison with AI-derived estimates.

The recent study was exploratory and based on a limited number of RTE meals; our analysis emphasized descriptive measures of variability and accuracy rather than formal hypothesis testing. Mean deviations and percentage deviations from labeled values were calculated, along with inter-dietitian coefficients of variation (CV%). This descriptive approach was chosen to present the extent and direction of estimation errors in a transparent manner. In addition, because food labels may vary even for the same type of RTE meal due to serving weight, preparation, or manufacturing tolerances, and since the study’s main objective was to examine the relative precision and variability of AI and dietitian assessments rather than absolute accuracy against labels, detailed reporting of individual label values was not emphasized.

This study does not involve human participants directly. However, ethical approval will be sought to ensure responsible use of food data and transparency when reporting AI decision processes. All RTE meals will be photographed in public retail contexts with no personal or proprietary data collected.

## 3. Results

This study evaluated eight RTE meals available in convenience stores, analyzing the accuracy of their nutrition labels. Four registered dietitians (1 male and 3 females; average age 36.2 ± 9.1 years; average clinical experience 6.1 ± 5.8 years) assessed the nutritional content of each meal based on their professional knowledge.

Table 1 lists the mean absolute coefficient of variation (CV%) for total calories, crude protein, crude fat, saturated fat, total carbohydrates, and sodium estimations of 4 dietitians over 8 RTEs were 5.6%, 14.3%, 33.4%, 24.5%, 11.1%, and 40.2%, respectively.

The variability of 5 AI models was calculated by estimating absolute CV% for total calories, crude protein, crude fat, saturated fat, total carbohydrates, dietary fiber, and sodium content of the RTE. Table 2 summarizes the intra-assay variation (Same Day, 3 Trials) of five AI chatbots when each model evaluated the same eight commercial meals three times on the same day. Inter-Assay Variation in AI Models (Table 3) shows the stability of the five AI chatbots, with three repeated assessments performed over three consecutive days using the same dialog prompts. CV% was calculated for the same set of nutrients. Table 4 and Figure 1 present a comparative analysis of the accuracy in estimating six major nutrients, calories, crude protein, crude fat, saturated fat, total carbohydrates, and sodium, by five AI chatbots and registered dietitians, using the actual labeled values of eight commercial meals as the reference.

## 4. Discussion

This study provides a novel comparative analysis of the accuracy and reproducibility of AI chatbot-generated nutrition assessments compared to professional dietitian estimates and commercial nutrition labels for RTE convenience meals. Among four registered dietitians, nutrient estimations showed low variation for calories, protein, and carbohydrates (CVs of 5.6%, 13.7%, and 11.0%, respectively), but higher variability for fat and sodium (CVs of 33.3% and 40.2%, respectively). These findings align with the prior literature, such as Braakhuis et al. in 2003, which suggests that professional experience and ingredient-level analysis enable stable assessments of macronutrients like calories and protein [12]. However, nutrients with variable content and preparation methods, such as fat and sodium, remain challenging to estimate even for experts, reflecting the influence of food preparation variability and hidden ingredients. The higher inter-rater coefficients of variation observed for fat, saturated fat, and sodium in our study likely arise from invisible factors such as hidden oils, brines, and sauces, as well as uncertainties in preparation (e.g., frying oil absorption), which may differ even among identical labeled products. These factors inevitably affect dietitians’ estimations. AI-assisted dietary assessment tools could help mitigate these limitations by reducing human bias in portion estimation and nutrient coding—a benefit particularly relevant in complex or mixed dishes, where recall bias and visual misclassification are common sources of error [13].

As AI applications in dietary counseling and food label analysis gain traction, understanding their limitations and reliability is critical for clinical and consumer use. Among tested models, Claude and ChatGPT exhibited the highest consistency, consistent with studies highlighting GPT-4’s strength in data reproducibility [1,2,14]. In contrast, Gemini showed significant variation in sodium estimates (CV 69.2%), and Grok displayed instability across multiple nutrients, including fat (31.1%) and carbohydrates (22.8%). These results indicate that some AI systems produce unstable outputs, limiting their reliability for precise clinical or dietary interventions. The study also assessed AI responses across three days using identical image prompts. ChatGPT and Claude maintained lower variability (CVs < 15% for most nutrients), while Gemini and Grok showed substantial inconsistencies, particularly in sodium and fat. This lack of stability highlights a key barrier to AI deployment in real-world dietary monitoring, where longitudinal consistency is essential for meal planning in chronic disease contexts like diabetes or chronic kidney disease (CKD).

Registered dietitians demonstrated the highest accuracy and consistency across all nutrient categories, with estimates closest to labeled values for calories (99.5%), protein (88.6%), and fat (88.3%). Even for challenging nutrients like saturated fat (87.4%) and sodium (91.8%), their results remained within acceptable error margins, underscoring the value of professional expertise and contextual knowledge. Among AI tools, ChatGPT showed balanced performance, accurately estimating protein (98.4%) and saturated fat (103.7%), though it overestimated fat (122.6%) and carbohydrates (136.3%). Claude and Gemini significantly overestimated fat (191.4% and 200.6%, respectively), despite moderate protein accuracy. Claude also underestimated carbohydrates (69.7%), indicating inconsistent performance. Grok overestimated calories (126.8%) and fat (145.2%) but was relatively accurate for protein (95.7%) and sodium (88.6%). Copilot performed poorly, underestimating carbohydrates (39.3%), saturated fat (35.1%), and protein (64.8%).

AI models exhibited varying accuracy but none consistently matched dietitians’ performance. While tools like ChatGPT offer reasonable estimates for some nutrients, they lack contextual sensitivity for nutrients like fat and sodium, which are heavily influenced by preparation methods. These findings emphasize the importance of human expertise in nutritional assessment and highlight AI’s current inability to replace professional judgment. Compared to nutrition labels, AI models showed larger deviations than dietitians. For instance, Gemini’s protein estimates deviated by over 70%, and Claude and ChatGPT’s fat estimates exceeded 40%, while dietitians’ deviations were only 14.3% for protein and 33.3% for fat. This aligns with prior reports, noting significant inaccuracies in AI and label data, raising concerns for health contexts requiring precision, such as carbohydrate counting for insulin therapy [1,7,14,15].

Figure 1 and Table 4 highlight the contrast between dietitians and AI chatbots in nutrient estimation accuracy. Dietitians consistently provided estimates closest to labeled values across six key nutrients: calories, protein, fat, saturated fat, carbohydrates, and sodium. Their performance reinforces the critical role of clinical experience, food preparation knowledge, and ingredient-level judgment, consistent with Braakhuis et al., which noted lower variability in macronutrient estimates but increased errors for nutrients affected by preparation, like fat and sodium [12]. Among the five AI models, ChatGPT performed acceptably, particularly for protein and saturated fat, though it overestimated fat and carbohydrates, aligning with Hoang et al. and Haman et al. further supported ChatGPT’s potential, noting 97% of caloric estimates were within ±40% of USDA values, with low variability, though errors persisted for fat and sodium due to cooking techniques and hidden ingredients [1,2].

However, from a clinical standpoint, consistent sodium underestimation by AI could lead to unintended excess intake in patients with hypertension, CKD, or heart failure—populations for whom sodium control is therapeutically vital [16]. This concern is particularly significant given the strong association between high sodium intake and increased risk of both hypertension and heart failure, as well as guidelines recommending sodium restriction in these patient groups [17]. Therefore, we emphasize that AI-generated estimates should serve only as preliminary guidance; high-stakes nutrients such as sodium should be verified by dietitians or, where feasible, directly measured.

Moreover, our study was conducted using RTE convenience meals from Taiwan, which are shaped by local culinary practices, ingredients, and labeling standards. These cultural and geographic factors limit the generalizability of our findings to other populations. For instance, differences in cooking methods, seasoning patterns, and the use of hidden oils or sauces between Asian meals and Western or Mediterranean cuisines may influence the accuracy of AI-based nutrient estimation. In addition, national food labeling regulations vary considerably, which may affect the reliability of reference values used for comparison [10,15]. Given that the consumption of ultra-processed foods differs widely across countries and regions, further validation of AI-assisted dietary assessment is needed in diverse food environments [18].

Several factors can contribute to inaccuracies in nutrient estimation. First, unpredictable elements like food processing techniques, frying oil absorption, hidden brines, and sauces introduce variability in fat and sodium content across seemingly similar products [19]. Second, AI systems face challenges when analyzing visually complex or occluded dishes; research shows that many image-based methods still struggle with portion-size estimation and ingredient recognition, especially in mixed meals [16]. Our study’s strengths include the use of multiple AI models evaluated through repeated measures (intra- and inter-assay) and blinded, independent assessments by registered dietitians—offering a robust comparison between human and machine performance.

Another limitation of our study is that we did not perform formal hypothesis testing or equivalence analysis, which could provide additional inferential insights. Instead, we focused on descriptive statistics (e.g., deviations and CV%) that directly convey the degree of variability across AI models and dietitian assessments. This choice reflects the exploratory nature of our work and the modest sample size; future studies with larger datasets may be able to incorporate more advanced statistical models.

Moreover, in this study, we did not analyze sugar, particularly the content of mono- and disaccharides. Although simple carbohydrates are important for nutritional health and disease risk, Taiwanese food labels report only total sugar, without further breakdown [20,21]. In addition, sugar estimation is inherently difficult in image-based and ingredient-level assessments because sugars are often ‘hidden’ in sauces, condiments, or processed components and cannot be visually separated [21,22]. Future research should specifically examine sugar estimation in AI-assisted dietary assessment, as this nutrient is highly relevant to metabolic health.

Finally, the results of this study should be interpreted as a snapshot in time, since AI models evolve rapidly and their performance can shift with each new version or update. This underscores the need for periodic re-validation of AI-based dietary assessment tools to ensure accuracy and reliability as technology advances [23]. From a practical perspective, AI-generated estimates may be most useful as a first step for consumers, providing quick guidance on energy and macronutrient content, while dietitians or clinicians verify critical nutrients such as sodium in populations where precision is essential [24]. For clinicians, AI tools could serve as a supplementary aid in patient education and dietary monitoring but should not replace professional judgment [25,26]. Improving the transparency of AI reasoning—for example, by making underlying assumptions about ingredients or portion sizes more explicit—will be crucial for building user trust and facilitating safe adoption in real-world dietary assessment [26].

## 5. Conclusions

In conclusion, while conversational AI chatbots show emerging utility in dietary analysis, their accuracy and consistency vary, particularly for protein, fat, and sodium. ChatGPT and Claude outperform others but deviate significantly from dietitian estimates and labels. Future improvements in AI design, nutrition-specific training, and visual recognition are essential for clinical use. Until then, AI nutritional information should complement, not replace, professional expertise. By estimating missing data, providing tailored feedback, and educating users accessibly, AI chatbots like ChatGPT enhance dietary awareness, support disease management, and elevate everyday food choices in modern life. Moreover, future research should extend the evaluation of AI-based nutrient estimation beyond convenience meals to include restaurant foods, home-prepared dishes, and condition-specific diets (e.g., for diabetes, chronic kidney disease, or heart failure), and studies with larger datasets should also apply more robust statistical frameworks to further validate AI performance across diverse dietary contexts and clinical populations.

## Figures and Tables

**Figure 1 nutrients-17-03044-f001:**
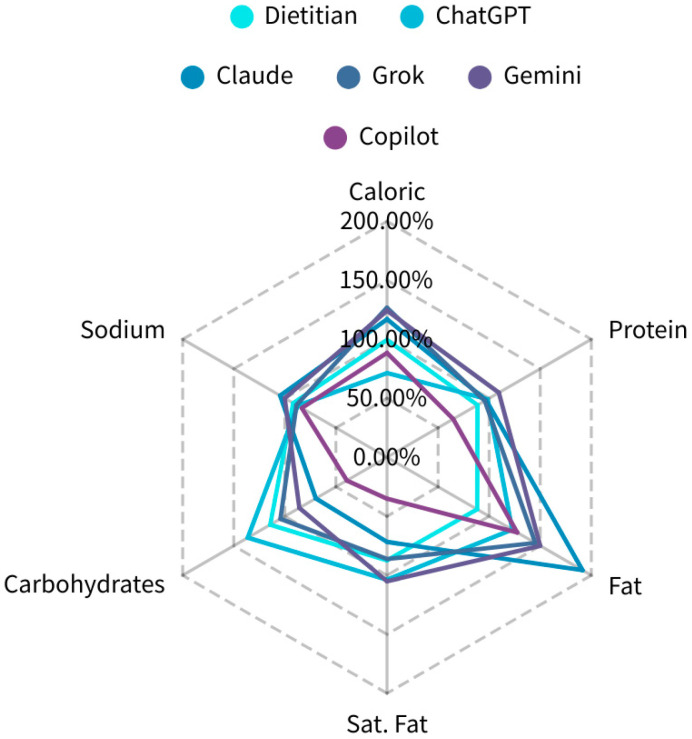
Radar plot comparing the accuracy of nutrient estimations for total energy, crude protein, crude fat, saturated fat, total carbohydrates, and sodium across five AI chatbots (ChatGPT, Claude, Grok, Gemini, Copilot) and dietitians. Values (dash lines) represent percentage deviation relative to labeled nutrient values.

**Table 1 nutrients-17-03044-t001:** Inter-Dietitian Variation. Summarizes the inter-dietitian variation (coefficient of variation) for total energy, crude protein, crude fat, saturated fat, total carbohydrates, and sodium estimated of 8 RTE.

Dietitian	Calory	Protein	Fat	Sat. Fat	Carbohydrates	Sodium
Dietitian 1	18.6 ± 11.2%	13.7 ± 11.7%	28.1 ± 18.0%	27.4 ± 17.7%	21.1 ± 10.8%	85.6 ± 53.0%
Dietitian 2	7.0 ± 4.5%	13.1 ± 9.0%	12.5 ± 9.1%	16.0 ± 11.7%	14.4 ± 7.5%	34.0 ± 36.0%
Dietitian 3	13.4 ± 6.2%	14.5 ± 7.3%	33.9 ± 26.6%	41.7 ± 25.9%	26.6 ± 6.7%	39.3 ± 32.2%
Dietitian 4	11.6 ± 7.1%	27.1 ± 17.1%	98.0 ± 165.4%	37.6 ± 25.2%	16.7 ± 6.7%	48.0 ± 32.2%
Mean	5.6 ± 3.4%	14.3 ± 6.0%	33.3 ± 37.6%	24.5 ± 11.7%	11.0 ± 6.8%	40.2 ± 30.3%

Data were expressed as mean ± SD.

**Table 2 nutrients-17-03044-t002:** Intra-Assay Variation in AI Models.

AI Model	Calories	Protein	Fat	Sat. Fat	Carbohydrates	Sodium
ChatGPT	8.8 ± 4.2%	11.8 ± 4.2%	14.0 ± 4.8%	14.7 ± 6.6%	9.5 ± 3.6%	34.6 ± 10.8%
Claude	5.5 ± 2.9%	9.0 ± 6.5%	20.4 ± 32.2%	18.2 ± 21.9%	20.4 ± 25.2%	21.7 ± 9.5%
Grok	16.7 ± 8.8%	13.6 ± 6.5%	31.1 ± 22.3%	28.3 ± 24.2%	22.8 ± 14.9%	33.1 ± 8.6%
Gemini	16.7 ± 3.1%	12.8 ± 6.9%	20.1 ± 9.5%	21.3 ± 10.6%	24.9 ± 8.2%	69.2 ± 10.9%
Copilot	10.9 ± 7.7%	10.2 ± 3.8%	16.8 ± 9.8%	27.7 ± 19.4%	22.0 ± 9.6%	28.3 ± 19.3%

Data were expressed as mean ± SD.

**Table 3 nutrients-17-03044-t003:** Inter-Assay Variation in AI Models.

AI Model	Calories	Protein	Fat	Sat. Fat	Carbohydrates	Sodium
ChatGPT	8.8 ± 4.2%	11.8 ± 4.1%	14.0 ± 4.8%	14.7 ± 6.6%	9.5 ± 3.6%	34.6 ± 10.8%
Claude	5.5 ± 2.9%	9.0 ± 6.5%	20.4 ± 32.2%	18.2 ± 21.9%	20.4 ± 25.2%	21.7 ± 9.5%
Grok	13.6 ± 7.2%	11.1 ± 5.3%	25.4 ± 18.2%	23.1 ± 19.7%	18.6 ± 12.2%	27.0 ± 7.0%
Gemini	13.6 ± 2.5%	10.5 ± 5.6%	16.4 ± 7.8%	17.4 ± 8.7%	20.3 ± 6.7%	56.5 ± 8.9%
Copilot	8.5 ± 6.3%	7.6 ± 3.7%	12.6 ± 7.9%	20.8 ± 17.0%	18.2 ± 7.7%	22.8 16.1%

Data were expressed as mean ± SD.

**Table 4 nutrients-17-03044-t004:** The accuracy for total energy, crude protein, crude fat, saturated fat, total carbohydrates, and sodium estimated results from five AI chatbots and dietitians against the actual nutrition labels of the eight commercial meals.

	Caloric	Protein	Fat	Sat. Fat	Carbohydrates	Sodium
Dietitian	99.5%	88.6%	88.3%	87.4%	114.3%	91.8%
ChatGPT	71.4%	98.4%	122.6%	103.7%	136.3%	87.4%
Claude	117.0%	97.0%	191.4%	71.7%	69.7%	104.6%
Grok	126.8%	95.7%	145.2%	86.2%	104.3%	88.6%
Gemini	124.2%	109.5%	200.6%	105.1%	85.8%	100.1%
Copilot	88.5%	64.8%	127.5%	35.1%	39.3%	83.4%

## Data Availability

The data that support the findings of this study are available on request from the corresponding authors, W.-C.Y. and W.-H.T., due to the absence of human subjects and the lack of ethical or confidentiality restrictions.

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
