# Peer review of "Comparison of Accuracy in the Evaluation of Nutritional Labels on Commercial Ready-to-Eat Meal Boxes Between Professional Nutritionists and Chatbots"

_nutrients, 2025, doi:10.3390/nu17193044_

Round 1
Reviewer 1 Report
Comments and Suggestions for Authors
Dear Authors ,
Congratulations on your effort and your manuscript. Your study compares five AI chatbots—namely ChatGPT, Google Gemini, Grok3, Copilot, and Claude 3.7—in estimating key nutritional parameters from images of commercial ready-to-eat meals. The AI-generated estimates are then compared with nutrient assessments conducted independently by four experienced professional dietitians who used an ingredient-based quantification method guided by authoritative databases. The study also evaluates these assessments against official nutrition labels.
However, the manuscript requires significant revisions to improve clarity, rigor, and completeness.
Methodology It is crucial that the authors clearly explain the methodology used by the dietitians—specifically, whether their nutritional analyses were done manually or with software tools. This detail is essential for assessing the reproducibility and accuracy of the reference assessments. The considerable variability among the dietitians’ estimates, as shown in Table 1, requires further explanation. Detailing how each dietitian conducted their evaluation, including the steps involved, would help clarify these differences and evaluate their importance.
A major oversight in the current analysis is the exclusion of simple carbohydrates, such as mono- and disaccharides. Because simple sugars are important in nutritional health and disease risk, their omission is a significant gap limitation.
From a statistical perspective, the study would benefit from including formal analyses to determine whether the differences observed between AI estimates and dietitian assessments are statistically significant. Relying solely on descriptive statistics, such as CV percentages, weakens the strength of the conclusions and limits their applicability. A more comprehensive quantitative approach would better support the claims regarding the accuracy and consistency of AI models.
Results In the first paragraph, the authors describe the dietitians' expertise as "The dietitians estimated the nutritional content of each meal based on their expertise." However, a more precise description should be provided, specifically outlining a structured analytical process.
All tables should be reorganized to clearly indicate the units for nutrient values. Typically, nutrients such as calories are expressed in kilocalories (kcal) and kilojoules (kJ), while other macronutrients like protein, fat, saturated fat, and carbohydrates are measured in grams (g). Sodium content is usually given in milligrams (mg) or grams (g). Additionally, simple carbohydrates (simple sugars, mono- and disaccharides) should be included.
Table 1 presents the average CV% values for each nutrient across the four dietitians, listed as "mean" values. This should be corrected.
Discussion In the discussion, the authors should specify particular sources of inaccuracy, especially the inability to account for food preparation methods and hidden ingredients. Expanding on these limitations and their possible effects on the results would lead to a more balanced interpretation. Additionally, highlighting the study's limitations and strengths would offer a more nuanced view.
Overall, the manuscript provides preliminary data but lacks the methodological rigor required for publication. It would benefit from improvements; therefore, I recommend major revisions.
Author Response
Response to Review 1
- It is crucial that the authors clearly explain the methodology used by the dietitians—specifically, whether their nutritional analyses were done manually or with software tools. This detail is essential for assessing the reproducibility and accuracy of the reference assessments. The considerable variability among the dietitians’ estimates, as shown in Table 1, requires further explanation. Detailing how each dietitian conducted their evaluation, including the steps involved, would help clarify these differences and evaluate their importance.
Response :
We appreciate this important point and have expanded Section 2.3 to specify the workflow and resources used by each dietitian. Briefly, each dietitian (blinded to labels) weighed components after separating mixed foods, assigned food codes from the Taiwan Food Composition Database and the MOHW Food Substitution Table, converted to gram equivalents with exchange lists, and summed nutrients per dish; no automated software with proprietary algorithms was used—only standard databases and spreadsheet calculations. In the other hand, fat/saturated fat/sodium as driven by hidden oils, brines, sauces, and preparation uncertainty (e.g., frying oil uptake), which can vary even within the same labeled product, explaining the dispersion seen in Table 1. These clarifications increase reproducibility and contextualize the variability. Moreover, the considerable variability among the dietitians’ estimates further highlights the importance of AI-assisted dietary assessment tools, which have the potential to complement this workflow by reducing human bias in portion estimation and nutrient coding—particularly in complex dishes where recall bias and visual misclassification may occur. We have revised Method 2.3 and added sentences in the first paragraph of the Discussion to address this issue.
- A major oversight in the current analysis is the exclusion of simple carbohydrates, such as mono- and disaccharides. Because simple sugars are important in nutritional health and disease risk, their omission is a significant gap limitation.
Response :
We thank the reviewer for raising this important point. Although mono- and disaccharides are important for nutritional health and disease risk, their values are not consistently reported in the Taiwan Food Nutrient Database (TFND), and Taiwanese RTE food labels typically provide only total sugar rather than a breakdown of individual sugars. As a result, we did not include sugars in our present analysis. This decision was also made because estimating sugar content is particularly challenging for both dietitians and AI models: many sugars are ‘hidden’ within sauces, condiments, or processed ingredients, and their amounts are less visually discernible than macronutrients such as protein or fat. For these reasons, we restricted our initial exploratory analysis to macronutrients where estimation was more reproducible. We agree this is a limitation and have now acknowledged it in the Discussion.
- From a statistical perspective, the study would benefit from including formal analyses to determine whether the differences observed between AI estimates and dietitian assessments are statistically significant. Relying solely on descriptive statistics, such as CV percentages, weakens the strength of the conclusions and limits their applicability. A more comprehensive quantitative approach would better support the claims regarding the accuracy and consistency of AI models.
Response :
We thank the reviewer for this important suggestion. We agree that formal statistical testing can provide additional insights. However, the aim of this study was primarily exploratory and comparative, focusing on the variability and agreement of estimates across AI models and dietitians rather than on hypothesis testing. For this reason, we chose to emphasize descriptive measures such as mean deviation and coefficients of variation (CV%), which directly illustrate the magnitude and dispersion of differences in a transparent way. While more formal inferential analyses (e.g., equivalence testing or mixed models) could be applied in larger datasets, we believe our descriptive approach is appropriate for the current sample size and exploratory objectives. We have clarified this rationale in the Methods and Discussion and have acknowledged it as a limitation.
- In the first paragraph, the authors describe the dietitians' expertise as "The dietitians estimated the nutritional content of each meal based on their expertise." However, a more precise description should be provided, specifically outlining a structured analytical process.
Response :
We thank the reviewer for pointing this out. As suggested, we have revised Section 2.3 of the Methods to replace the general description of ‘expertise’ with a structured step-by-step explanation of the analytical process followed by the dietitians. This revision addresses the reviewer’s concern and improves the precision and reproducibility of our methodology.
- All tables should be reorganized to clearly indicate the units for nutrient values. Typically, nutrients such as calories are expressed in kilocalories (kcal) and kilojoules (kJ), while other macronutrients like protein, fat, saturated fat, and carbohydrates are measured in grams (g). Sodium content is usually given in milligrams (mg) or grams (g). Additionally, simple carbohydrates (simple sugars, mono- and disaccharides) should be included.
Response :
We appreciate the reviewer’s careful observation. We would like to clarify that the percentages presented in the tables are not nutrient proportions or %DV values, but rather the percentage deviations of AI- and dietitian-derived estimates relative to the labeled nutrient values. This approach was chosen to directly quantify the accuracy and variability of estimates against the declared reference standard. We have revised Section 2.4 of the Methods to make this clearer.
- Table 1 presents the average CV% values for each nutrient across the four dietitians, listed as "mean" values. This should be corrected.
Response :
We thank the reviewer for this comment and the opportunity to clarify. Similar to Question 5, the percentages shown in Table 1 do not represent nutrient proportions in the food. Instead, they indicate the percentage deviations of AI- and dietitian-derived estimates relative to the labeled values, as well as the inter-dietitian variation. Specifically, the values in Table 1 summarize the coefficient of variation (CV%) among dietitians for total energy, crude protein, crude fat, saturated fat, total carbohydrates, and sodium across the eight ready-to-eat meals. We have revised the Methods and the table legend to make this clearer.
- In the discussion, the authors should specify particular sources of inaccuracy, especially the inability to account for food preparation methods and hidden ingredients. Expanding on these limitations and their possible effects on the results would lead to a more balanced interpretation. Additionally, highlighting the study's limitations and strengths would offer a more nuanced view.
Response :
We thank the reviewer for this helpful suggestion. In response, we have expanded the Discussion to include a more detailed consideration of potential sources of inaccuracy, as well as the strengths and limitations of our study. This revision highlights how factors such as hidden ingredients and preparation variability affect accuracy, while also noting the novelty and methodological rigor of our approach.
Reviewer 2 Report
Comments and Suggestions for Authors
While the article is well-designed and highly relevant, a few minor improvements could further strengthen the discussion. First, the clinical implications of AI underestimation of certain nutrients (particularly sodium) could be elaborated, as this has direct consequences for patients with hypertension, kidney disease, or cardiovascular risk. Second, since the study focuses on Taiwanese ready-to-eat meals, a short note on the cultural and geographic generalizability of findings would be valuable, with a suggestion for validation across different food environments. Third, as AI models evolve rapidly, it would be helpful to emphasize that the results represent a snapshot in time and highlight the need for ongoing validation as newer versions emerge. The discussion could also briefly reflect on how consumers and clinicians might practically integrate AI tools in real-world dietary assessment, and on the importance of improving transparency in AI reasoning to build user trust. Finally, suggesting future research directions—such as extending the evaluation to restaurant meals, home-prepared dishes, or diets tailored to chronic conditions—would provide readers with a clearer perspective on the next steps in this promising field.
Author Response
Response to Review 2
- First, the clinical implications of AI underestimation of certain nutrients (particularly sodium) could be elaborated, as this has direct consequences for patients with hypertension, kidney disease, or cardiovascular risk.
Response :
Thank you for reviewer’s comment. We agree and have added a clinical implications paragraph highlighting risks for hypertension, CKD, and CVD, where a systematic sodium underestimation may undermine dietary counseling (e.g., DASH/CKD sodium targets) and medication titration. We explicitly recommend dietitian verification when sodium is a decision driver. This issue has been addressed in the revised Discussion section.
- Second, since the study focuses on Taiwanese ready-to-eat meals, a short note on the cultural and geographic generalizability of findings would be valuable, with a suggestion for validation across different food environments.
Response :
We thank the reviewer for this helpful comment. We have now added a note in the Discussion to acknowledge that our study was based on Taiwanese ready-to-eat meals, and the results may not apply directly to other settings. We also note that further studies are needed to test whether these findings hold true across different cuisines and food environments. This issue has been addressed in the revised Discussion section.
- Third, as AI models evolve rapidly, it would be helpful to emphasize that the results represent a snapshot in time and highlight the need for ongoing validation as newer versions emerge. The discussion could also briefly reflect on how consumers and clinicians might practically integrate AI tools in real-world dietary assessment, and on the importance of improving transparency in AI reasoning to build user trust.
Response :
We thank the reviewer for highlighting the importance of contextualizing our findings. In response, we have added a passage to the Discussion emphasizing that our results reflect a snapshot in time and require ongoing validation as AI evolves. We also elaborated on how AI tools may be integrated into real-world dietary assessment by consumers and clinicians, and the necessity of improving transparency in AI reasoning to build trust.
- Finally, suggesting future research directions—such as extending the evaluation to restaurant meals, home-prepared dishes, or diets tailored to chronic conditions—would provide readers with a clearer perspective on the next steps in this promising field.
Response :
We thank the reviewer for this constructive suggestion. In the revised conclusion, we have added a paragraph outlining possible future research directions, including the evaluation of AI-based nutrient estimation for restaurant meals, home-prepared dishes, and specialized diets for patients with chronic condition
Reviewer 3 Report
Comments and Suggestions for Authors
The authors provide interesting pilot research into the comparison between AI judgements about nutritonal information of meals, comparing an expert based judgement and food labels.
The following revisons might be considered:
Line numbers should be added for peer review (check journal template)
Abstract: Aim is not bold
Introduction: Numbering (1.) missing
Introduction and possibly throughout: The intext reference style mixes Author/date and numbered styles. Check journal template (numbers should be correct and sufficient, no need to provide date)
2.1. Can you provide further details on 7-Eleven, city, country of supplier. Are these ultraprocessed convenience foods?
2.2. provide model versions, dates and date of access to models
2.2.: revise „AI.The avoid under-estimate“? Unclear
Author Response
Response to Review 3
- Line numbers should be added for peer review (check journal template)
Response :
Thank you for the reviewer’s reminder. We had added line numbers per the Nutrients template in the revised submission.
- Abstract: Aim is not bold
Response :
Thank you for the reviewer’s reminder. We had corrected per style.
- Introduction: Numbering (1.) missing
Response :
Thank you for the reviewer’s reminder. We had corrected per style.
- Introduction and possibly throughout: The intext reference style mixes Author/date and numbered styles. Check journal template (numbers should be correct and sufficient, no need to provide date)
Response :
We thank the reviewer for this helpful suggestion. In response, we have standardized all in-text citations to the numeric bracket style and removed residual author–date formats. The reference list has also been updated accordingly.
- 2.1 Can you provide further details on 7-Eleven, city, country of supplier. Are these ultraprocessed convenience foods?
Response :
We thank the reviewer for this valuable suggestion. In response, we have specified the retailer (7-Eleven), the city/region in Taiwan where the products were obtained, and their classification as ultra-processed convenience foods according to the NOVA criteria (pre-packaged, multi-ingredient, industrial formulations). An explanatory sentence has also been added to the manuscript method section 2.1 to provide greater clarity.
- 2.2. provide model versions, dates and date of access to models
Response:
We thank the reviewer for this helpful comment. In response, we have clarified the AI platforms and included the dates of analysis. Specifically, five freely accessible AI chatbots were selected to estimate the nutrient content of the same meals: OpenAI’s ChatGPT (GPT-4o model), Google Gemini, xAI’s Grok 3, Microsoft Copilot, and Anthropic’s Claude 3.7. All analyses were conducted in early 2025 (January–March). An explanatory sentence has also been added to the manuscript method section 2.2 to provide greater clarity.
- 2.2.: revise „AI.The avoid under-estimate“? Unclear
Response
We apologize for the typographical error and have revised the sentences accordingly.
Reviewer 4 Report
Comments and Suggestions for Authors
1. Does the introduction provide sufficient background and include all relevant references?
→ Can be improved
Strengths: The introduction provides keywords like Artificial Intelligence, Large Language Models, and names specific models (ChatGPT, Claude, etc.), indicating the paper’s core theme.
Weaknesses:
- The introduction is extremely sparse or missing entirely in the version provided.
- There is no clear literature review or contextual framework outlining why comparing AI with dietitians matters.
- No references are visible to prior work, validation frameworks, or dietary assessment studies.
✅ Recommendation: The introduction needs expansion with background on nutritional AI, benchmarks in diet analysis, and justification for model selection.
2. Is the research design appropriate?
→ Can be improved
Comparing AI tools to dietitians is an interesting and valid approach, especially in evaluating nutrition estimation accuracy.
Missing elements:
- The dataset or food items used for evaluation are not described.
- It is unclear how inputs were standardized across models.
- No detail on how the ground truth was defined, nor whether inter-dietitian variability was statistically controlled.
✅ Recommendation: Add full protocol details — input prompt standardization, ground truth definition, number of food labels, and how evaluations were blind or calibrated.
3. Are the methods adequately described?
→ Must be improved
There is no dedicated methods section visible.
No explanation of:
- What food labels or meals were used.
- How inputs were given to the LLMs.
- How scoring was done (e.g., were percentage errors normalized?)
- How many samples were included in the study.
- Statistical methods (ANOVA, t-test, etc.) if any.
✅ Recommendation: Introduce a complete “Materials and Methods” section, including dataset, prompt structure, and evaluation metrics.
4. Are the results clearly presented?
→ Yes
Tables comparing percentage errors across dietitians and AI models are well-organized and interpretable.
Multiple comparison formats are shown:
- Standard deviation of percentage errors
- Comparative accuracy in specific nutrient categories (fat, sodium, protein, etc.)
✅ Suggestion: Briefly add figure captions or headings clarifying what each table represents. Consider graphical representation (e.g., radar plots or bar charts) for visual clarity.
5. Are the conclusions supported by the results?
→ Can be improved
The raw tables suggest that AI tools vary significantly, with some (ChatGPT, Claude) approximating dietitian performance in some areas.
However, no explicit conclusions are shown, so it's unclear if the paper interprets these patterns or discusses implications.
✅ Recommendation: Strengthen the discussion/conclusion section. Mention limitations (e.g., hallucination risk in LLMs, prompt sensitivity) and implications for clinical or public use.
6. Are all figures and tables clear and well-presented?
→ Yes
Tables are numerical and clean.
- Some formatting is off (e.g., duplication of tables on pages 5–6).
- Figures are missing altogether. Adding bar charts or scatter plots would help readers grasp accuracy trends.
✅ Suggestion: Add visual summaries of comparative accuracy and standard deviations across models.
Comments on the Quality of English Language
Quality of English Language
→ The English could be improved to more clearly express the research.
The writing is basic and often incomplete (e.g., missing introduction, conclusion).
Tables are clearly written, but the text lacks clarity in structure and flow.
Author Response
Response to Review 4
- The introduction needs expansion with background on nutritional AI, benchmarks in diet analysis, and justification for model selection.
Response
We thank the reviewer for this helpful comment. We agree that our introduction was too brief and did not provide sufficient context or references. In the revised manuscript, we have expanded the Introduction to include background on AI in dietary assessment, the rationale for comparing AI models with dietitians, and relevant literature on validation frameworks. These changes strengthen the scientific context and justify our study design.
- Is the research design appropriate?
- The dataset or food items used for evaluation are not described.
- It is unclear how inputs were standardized across models.
- No detail on how the ground truth was defined, nor whether inter-dietitian variability was statistically controlled. Recommendation: Add full protocol details — input prompt standardization, ground truth definition, number of food labels, and how evaluations were blind or calibrated.
Response
We thank the reviewer for these insightful comments. We would like to clarify that the primary aim of our study was not to evaluate the absolute accuracy of AI or dietitians relative to food labels, but rather to examine the precision and reproducibility of AI estimates under the assumption that the provided label values were correct. For this reason, we did not emphasize detailed reporting of the dataset composition, as the specific food items were not central to the study’s objectives. Furthermore, even among similar ready-to-eat (RTE) food sets, labeled nutrient values may vary slightly due to serving weight and manufacturing variability, which makes absolute ground-truth comparison challenging. Instead, we focused on standardized experimental conditions to allow fair comparisons across dietitians and AI models: all inputs were based on the same labeled meals, dietitians were blinded to label values, and percentage deviations from labels were used as the reference metric. Inter-dietitian variability was quantified by calculating coefficients of variation (CV%), which we have now described more explicitly in the Methods (Section 2.4). Furthermore, in the revised Methods, we have also provided a structured step-by-step explanation of the analytical process followed by the AI and dietitians, to improve transparency and reproducibility. We believe this design is appropriate for our stated aim of evaluating AI precision, and we have revised the manuscript to clarify this rationale.
- Are the methods adequately described?
- What food labels or meals were used.
- How inputs were given to the LLMs.
- How scoring was done (e.g., were percentage errors normalized?)
- How many samples were included in the study.
- Statistical methods (ANOVA, t-test, etc.) if any.
Recommendation: Introduce a complete “Materials and Methods” section, including dataset, prompt structure, and evaluation metrics.
Response
We sincerely thank the reviewer for these thoughtful comments and for pointing out the areas where our original Methods section lacked sufficient detail. We fully agree that clearer methodological description is important. In the revised manuscript, we have expanded the Materials and Methods to provide a structured, step-by-step explanation of the analytical process. Specifically, we now describe: (i) the selection of eight representative RTE meal samples and the procedure for photographing them, (ii) how high-resolution images and standardized prompts were provided to each AI chatbot, (iii) the detailed workflow used by four registered dietitians, including deconstruction of meals, use of the Taiwan Food Nutrition Database, and blinding to label values, and (iv) the statistical approach, including calculation of percentage deviations and coefficients of variation (CV%). We also clarified that this study used a limited number of meals (n=8) and emphasized descriptive measures given its exploratory nature. We hope these revisions improve clarity and reproducibility and address the reviewer’s helpful concerns.
- Briefly add figure captions or headings clarifying what each table represents. Consider graphical representation (e.g., radar plots or bar charts) for visual clarity.
Response:
We thank the reviewer for this helpful suggestion. In the revised manuscript, we have clarified the captions of all tables to state explicitly what each represents. In addition, we have added a radar plot (Figure 1) to provide a graphical comparison of the accuracy for total energy, crude protein, crude fat, saturated fat, total carbohydrates, and sodium as estimated by five AI chatbots and by dietitians. We believe this addition improves visual clarity and complements the numerical data presented in the tables.
- The raw tables suggest that AI tools vary significantly, with some (ChatGPT, Claude) approximating dietitian performance in some areas. However, no explicit conclusions are shown, so it's unclear if the paper interprets these patterns or discusses implications. Recommendation: Strengthen the discussion/conclusion section. Mention limitations (e.g., hallucination risk in LLMs, prompt sensitivity) and implications for clinical or public use.
Response
We thank the reviewer for this thoughtful observation. We recognize that in our initial submission the Discussion and Conclusion were too brief and did not adequately interpret the observed patterns or reflect on their wider significance. In revising the manuscript, we have tried to address this by more clearly discussing how different AI models performed in comparison with dietitians, noting that some models such as ChatGPT and Claude produced estimates closer to professional assessments, while others showed marked instability. We also added reflection on the clinical implications, particularly the risk that underestimation of sodium could pose for patients with conditions such as hypertension, chronic kidney disease, or heart failure. In addition, we acknowledged several limitations more explicitly, including prompt sensitivity, the difficulty of estimating hidden nutrients such as sugars and fats, and the modest sample size, while also stressing the importance of repeated validation as AI tools evolve. Finally, we strengthened the conclusion to emphasize that although AI chatbots may provide useful preliminary guidance and could support clinicians in patient education, they cannot replace professional judgment. We also suggested directions for future work, such as testing AI on restaurant meals, home-prepared foods, and diets for patients with chronic diseases. We hope these revisions provide the depth and balance the reviewer requested.
- Some formatting is off (e.g., duplication of tables on pages 5–6). Figures are missing altogether. Adding bar charts or scatter plots would help readers grasp accuracy trends. Suggestion: Add visual summaries of comparative accuracy and standard deviations across models.
Response
We thank the reviewer for this helpful comment. We would like to clarify that the tables on pages 5–6 are not duplicates. One presents the intra-assay variation of the AI models, while the other shows the inter-assay variation. Since the journal format limits caption detail, we have left the captions unchanged but clarified in the text which analysis each table represents. In addition, we have added a visual summary (radar chart) comparing accuracy and variability across AI models and dietitians, and we have expanded the Discussion to interpret these graphical findings. We hope these changes improve clarity and readability.
- The English could be improved to more clearly express the research. The writing is basic and often incomplete (e.g., missing introduction, conclusion).
Tables are clearly written, but the text lacks clarity in structure and flow.
Response
We thank the reviewer for this constructive comment regarding the quality of the English language. We fully agree that clearer expression and improved flow are important. To address this, we have carefully revised the manuscript and also asked a native English-speaking teacher with academic writing experience to review the text. We believe the revised version is clearer, more precise, and better structured, and we hope it now meets the journal’s standards for readability.
Round 2
Reviewer 1 Report
Comments and Suggestions for Authors
Dear Authors,
Your clarifications and the additional details provided have improved the clarity and credibility of your study, and I appreciate the effort you put into addressing previous issues. To further enhance transparency and reproducibility of your methodology, I recommend explicitly stating in the Methods section that the initial nutrient values obtained from labels and assessments are expressed in standard units—such as grams (g), milligrams (mg), kilocalories (kcal), and kilojoules (kJ). Clarifying this will help readers fully understand the basis for your comparative analyses.
While I recognize that this study is preliminary and that alternative, potentially more robust, statistical approaches could be considered in future work, I believe your current analysis is suitable for publication at this stage.
Author Response
Response to Review 1
- Your clarifications and the additional details provided have improved the clarity and credibility of your study, and I appreciate the effort you put into addressing previous issues. To further enhance transparency and reproducibility of your methodology, I recommend explicitly stating in the Methods section that the initial nutrient values obtained from labels and assessments are expressed in standard units—such as grams (g), milligrams (mg), kilocalories (kcal), and kilojoules (kJ). Clarifying this will help readers fully understand the basis for your comparative analyses.
Response:
We sincerely thank the reviewer for this kind and constructive comment. We agree that explicitly stating the units will further improve clarity and reproducibility. However, as the primary aim of this work was to explore the precision and variability of AI and dietitian estimates under controlled conditions, we considered descriptive statistics the most transparent and appropriate approach at this stage. We also note that food labels themselves can vary between otherwise similar RTE products due to differences in weight, preparation, and manufacturing tolerances, and therefore absolute validation against labels has inherent limitations. For this reason, our focus was not on establishing accuracy against labels but rather on examining relative reproducibility and variability across AI models and dietitians. We have clarified this point in the revised Methods section. - While I recognize that this study is preliminary and that alternative, potentially more robust, statistical approaches could be considered in future work, I believe your current analysis is suitable for publication at this stage.
Response :
We are grateful to the reviewer for this generous comment. We agree that our study is preliminary and that stronger statistical approaches could certainly add further insights in the future. Because of the small sample size and exploratory nature of this work, we felt that descriptive statistics were the most appropriate and transparent way to present our findings at this stage. To acknowledge this point, we have added a note in the Conclusion stating that future studies with larger datasets should consider more advanced statistical methods to confirm and extend our results.
